# Whole-genome sequencing of *Burkholderia glumae* strains from Thailand reveals potential horizontal gene transfer with *Burkholderia pseudomallei*

Sujin Patarapuwadol[1], Woranich Hintong[2,3], Pornpavee Nualnisachol[2,3], Natnicha Wankaew[4,5], Worarat Kruasuwan[4,5], Thanchanok Sawaengwong[4,5], Phatcharin Laosena[6], Jiraphan Premsuriya[2,3]*

**1** Department of Plant Pathology, Faculty of Agriculture at Kamphaeng Saen, Kasetsart University Kamphaeng Saen Campus, Nakhon Pathom, Thailand, **2** Princess Srisavangavadhana Faculty of Medicine, Chulabhorn Royal Academy, Bangkok, Thailand, **3** Research Center on Clinical and System Microbiology (RCSym), Chulabhorn Royal Academy, Bangkok, Thailand, **4** Division of Medical Bioinformatics, Research Department, Faculty of Medicine, Siriraj Hospital, Mahidol University, Bangkok, Thailand, **5** Siriraj Long-Read Lab (Si-LoL), Faculty of Medicine, Siriraj Hospital, Mahidol University, Bangkok, Thailand, **6** Program in Applied Biological Sciences, Chulabhorn Graduate Institute, Bangkok, Thailand

* jiraphan.pre@cra.ac.th

## Abstract

*Burkholderia glumae* is an emerging phytopathogen that causes bacterial panicle blight in rice and has been implicated in rare human infections. In Thailand, *B. glumae* and the human pathogen *Burkholderia pseudomallei* coexist in rice fields. Given the high genomic plasticity of *Burkholderia* species, including frequent genome rearrangements, variability in mobile genetic elements, and recombination events that facilitate horizontal gene transfer, there are concerns about the emergence of novel traits that may affect both plant and human health. In this study, we performed whole-genome sequencing and a comparative genomic analysis of 16 *B. glumae* strains isolated from rice fields across seven Thai provinces. Our phylogenomic analysis, based on core-genome single-nucleotide polymorphisms, revealed high genetic diversity and a polyclonal population structure, with evidence of a globally distributed clonal lineage. All isolates harbored plasmids and diverse prophage elements, which indicated extensive mobilome variability. A total of 572 putative horizontally transferred genes were identified. Most of these genes originated from unclassified or plant-associated *Burkholderia* species. Notably, two strains shared a chromosomal island that carried genes that were very similar to those found in *B. pseudomallei*. This genomic region contained genes associated with mobile genetic elements, phage defense, and a type VI secretion system, including genes that encode a PAAR domain–containing protein, a putative nuclease, and an immunity protein. Our findings highlight the genomic heterogeneity of *B. glumae* in Thailand and provide

**Data availability statement:** The WGS data obtained from the isolates analyzed in this study are available under BioProject accession number PRJNA1307573 in the NCBI database.

**Funding:** This research was supported by Chulabhorn Royal Academy (https://www.cra.ac.th/). Fundamental Fund: fiscal year 2024 by National Science Research and Innovation Fund, grant number 198501. The funder had no role in study design, data collection and analysis, decision to publish, or preparation of the manuscript.

**Competing interests:** The authors have declared that no competing interests exist.

evidence of interspecies horizontal gene acquisition from human pathogenic *B. pseudomallei*. The presence of *B. pseudomallei*-derived genes in *B. glumae* chromosomes underscores the potential for genetic exchange in shared environmental niches, which could affect the evolutionary dynamics and pathogenicity of *B. glumae*. Hence, our findings also emphasize the critical need for environmental surveillance and genome-based monitoring to track emerging genomic combinations relevant to both plant and human health.

## Introduction

*Burkholderia glumae* is a Gram-negative, rod-shaped bacterium that causes bacterial panicle blight (BPB) in rice. This seed-borne phytopathogen has the potential to cause yield losses of up to 75% in severely affected rice fields [1]. BPB was first documented in Japan during the 1950s and has since emerged as a significant threat to regional rice production in Asia, Africa, and the Americas [1]. This pathogen is predominantly distributed in tropical and subtropical regions, where environmental conditions are favorable for its proliferation. Climate change is likely to positively affect the spread of *B. glumae*, which may lead to increases in the incidence and severity of *B. glumae* infections in rice-producing regions [2]. In terms of its capacity to infect humans, *B. glumae* was isolated from a patient with chronic granulomatous disease [3].

In Thailand, *B. glumae* is recognized as an emerging rice pathogen. It was first isolated in 2011, and its identity was confirmed in 2017 [4]. Thailand is also an endemic region for melioidosis, a life-threatening disease caused by *Burkholderia pseudomallei*. This highly pathogenic bacterium is commonly found in the environment, particularly in rice fields [5]. Given that *B. glumae* and *B. pseudomallei* share the environmental niche of rice fields, it is highly likely that they coexist in Thai rice paddies and interact directly. In addition, the prevalence of both species increases with higher humidity and rainfall [2,6]. As a result, the population densities of both organisms are likely to peak simultaneously during the high-precipitation rice-growing season. The coincident population dynamics of *B. glumae* and *B. pseudomallei* may intensify their niche sharing, resulting in frequent cell-to-cell contact within the highly active and dense microbial communities of waterlogged rice-field soils. These high-density and environmentally stressed conditions provide an optimal setting for genetic exchange, thereby increasing the likelihood of horizontal gene transfer (HGT) mediated by mobile genetic elements or bacteriophages [7,8].

Members of the *Burkholderia* genus exhibit remarkable genomic plasticity driven by genome rearrangements, recombination events, and the acquisition of diverse mobile genetic elements such as plasmids, integrative elements, and prophages, which facilitate HGT within and between *Burkholderia* species [9,10]. For example, *Burkholderia thailandensis* and *Burkholderia cepacia* complex members have been shown to acquire and express *B. pseudomallei* capsular biosynthesis genes [11–13]. Furthermore, *B. glumae* has demonstrated the capacity to acquire genes from

other bacterial taxa, including *Pseudomonas* and *Streptomyces* species, which highlights its potential for acquiring novel genetic material through HGT and potentially increase its adaptability and virulence [14,15]. Hence, HGT plays an important role in shaping the virulence, environmental adaptation, and antimicrobial resistance of *Burkholderia* species [10]. The coexistence of *B. glumae* and *B. pseudomallei* in Thai environments, coupled with extensive evidence of horizontal gene transfer within *Burkholderia*, makes it critical to determine whether these two species exchange genes in nature. However, no study to date has examined genomic evidence of interspecies HGT in *B. glumae* isolates from Thailand. Addressing this gap is essential, as such gene exchange may influence the evolution of virulence, environmental adaptation, and the broader pathogenic landscape with implications for both agriculture and public health.

In this study, we performed whole-genome sequencing (WGS) and a comparative genomic analysis of 16 *B. glumae* isolates from Thailand to investigate evidence of HGT events between *B. glumae* and other *Burkholderia* species, particularly *B. pseudomallei*. WGS combined with comparative genomics allows high-resolution detection of mobile genetic elements, horizontally acquired regions, and evolutionary relationships across species. This approach is therefore well suited for identifying HGT events and elucidating genome evolution in *Burkholderia* [14,15]. Our findings provide new insights into the evolutionary dynamics of *B. glumae* and contribute to a better understanding of its genomic relationships with clinically important *Burkholderia* species.

## Materials and methods

### *B. glumae* strains

*B. glumae* strains isolated from rice fields in seven Thai provinces between 2011 and 2017 were obtained from the Department of Plant Pathology, Faculty of Agriculture at Kamphaeng Saen, Kasetsart University, Kamphaeng Saen Campus, Thailand. Bacterial isolation and identification were performed as previously described [4,16]. Bacterial stock cultures were stored in 20% glycerol at −80°C until use. This study did not involve new field sampling. All isolates were obtained from previously published collections, and no additional permits were required. Ethics approval was waived as no personal identifiers were included.

### DNA extraction and WGS

The *B. glumae* strains were streaked onto nutrient agar plates and incubated at 35°C overnight. Total DNA was extracted using Genomic DNA Purification Kits (Zymo Research, USA) following the manufacturer's protocol, with slight modifications. Bacterial colonies were collected from the entire agar surface rather than from a liquid culture, and the bead-beating step was shortened to 3 min [17]. DNA concentration and purity were assessed using a NanoDrop spectrophotometer (Thermo Fisher Scientific, USA). The purified DNA was aliquoted for both short-read and long-read sequencing. Short-read sequencing was performed by Novogene (Singapore) using the Illumina NovaSeq 6000 platform with 150 bp paired-end reads. For long-read sequencing, libraries were prepared from 50 ng of input DNA using the Rapid Sequencing gDNA – Barcoding Kit (SQK-RBK114.96; Oxford Nanopore Technologies [ONT], UK) and sequenced using a PromethION P24 device and an R10.4.1 flow cell (ONT, UK) under default settings. Base calling and quality control were performed using Dorado v0.7.3 with the SUP model v5.0.0 [18].

### Bioinformatic analysis

The quality of the Illumina short reads was assessed using FastQC v0.11.9 [19]. Adapter sequences were trimmed; reads with a quality score ≤ Q30 were filtered out using Fastp v0.23.2 [20]. For the ONT data, the raw reads were checked for their quality, trimmed with Porechop v0.2.4 (https://github.com/rrwick/Porechop), and filtered using Filtlong v0.2.1 (https://github.com/rrwick/Filtlong) to retain reads longer than 1,000 bp, with a minimum quality score > Q10. The quality of the filtered ONT reads was further assessed using NanoPlot v1.38.0 [18]. Hybrid genome assembly was performed using

Unicycler v0.4.8 [21]. Assembly completeness and contamination levels were evaluated using CheckM v1.2.1 [22]. Assembly statistics and genome metrics were analyzed using QUAST v5.0.2 [23]. Bacterial chromosomes were identified by aligning contigs with all marker genes in the GTDB-Tk database (R214)[24], and the average nucleotide identity (ANI) and alignment fraction were calculated using GTDB-Tk v2.1.1 [24]. Genome annotation was performed using Prokka V1.14.6 [25].

Phylogenetic analysis was performed using 16 newly sequenced genomes and 36 previously published *B. glumae* genomes available in the GenBank database (accessed on 5 June 2025). Only genomes with assembly status at the scaffold level or higher were included in the analysis. Core-genome single-nucleotide polymorphisms (SNPs), including substitutions, deletions, and insertions, were identified using Snippy v4.6.0 (https://github.com/tseemann/snippy). A maximum-likelihood phylogenetic tree based on the SNPs was constructed using IQ-tree v2.4.0 [26], with the GTR+ G model and 1,000 bootstrap replicates and *B. glumae* BGR1 as the reference genome (NC_012724.2 and NC_012721.2 for chromosomes 1 and 2, respectively). The final tree was visualized using iTOL v6 [27].

Plasmid typing was performed using MOB-suite v3.1.5 [28] and PlasFlow [29]. Prophage regions were predicted using PHASTER [30]. Putative horizontally transferred genes (HTGs) were identified by performing BLAST searches of the Horizontal Gene Transfer Database (HGT-DB) [31] using cutoff values of ≥95% for both sequence identity and query coverage. Candidate HTGs were subsequently validated through additional BLAST searches of the NCBI nonredundant database [32]. Horizontal transfer regions on the chromosome were predicted using Alien Hunter v1.3.0 [33], and the genomic regions containing putative HTGs were aligned and visualized using PyGenomeViz (https://github.com/moshi4/pyGenomeViz).

## Results

### Study strains and their general genomic features

A total of 16 *B. glumae* strains isolated from rice plants showing panicle blight disease in seven Thai provinces between 2011 and 2017 were included in this study (Table 1). The genomic sequences of 60BGCRPA10-1, 60BGCRMSO3-11, 60BGCRMSO3-9, 60BGCRMSO1-5, 60BGCRWC8-5, and 60BGCRMSO3-5 were sequenced and reported previously

**Table 1. Isolation information and sequencing status of *B. glumae* strains analyzed in this study.**

| Isolate | Province | Year | Rice cultivar | Sequencing status |
|---|---|---|---|---|
| 60BGCRPA10-1 | Chiang Rai | 2017 | Japonica rice | Re-sequenced |
| 60BGCRMSO3-11 | Chiang Rai | 2017 | Japonica rice | Re-sequenced |
| 60BGCRMSO3-9 | Chiang Rai | 2017 | Japonica rice | Re-sequenced |
| 60BGCRMSO1-5 | Chiang Rai | 2017 | Japonica rice | Re-sequenced |
| 60BGCRWC8--5 | Chiang Rai | 2017 | Japonica rice | Re-sequenced |
| 60BGCRMSO3-5 | Chiang Rai | 2017 | Japonica rice | Re-sequenced |
| 3BGNP2-3 | Nakhon Pathom | 2013 | Suphan Buri 1 | De novo |
| 3BGNY1-1 | Nakon Nayok | 2013 | RD47 | De novo |
| 1BGCR2-2 | Chiang Rai | 2012 | Japonica rice | De novo |
| 3BGST2-1 | Sukhothai | 2013 | Phitsanulok 2 | De novo |
| 1BGSP3-4 | Suphan Buri | 2012 | Phitsanulok 2 | De novo |
| 2BGCN4-1 | Chai Nat | 2013 | RD41 | De novo |
| 3BGNP1-4 | Nakhon Pathom | 2013 | Suphan Buri 1 | De novo |
| 3BGNY2-4 | Nakhon Nayok | 2013 | RD47 | De novo |
| 1BGRE5-1 | Roi Et | 2011 | KDML105 | De novo |
| 2BGCN5-2 | Chai Nat | 2013 | RD47 | De novo |

using Illumina sequencing [16]. In this study, the genomes of these strains were re-sequenced using both Illumina and ONT to ensure the inclusion of high-quality complete genomes. The WGS and assembly of the 16 *B. glumae* strains revealed high-quality genomes, with all samples assembled into two complete chromosomal contigs with sizes of 3.5–3.7 Mb and 2.7–2.9 Mb for chromosomes 1 and 2, respectively. The GC content was consistently around 68%, and the completeness score was 99.95% for all genomes. Taxonomic analysis confirmed all isolates as *B. glumae*, with ANI values of 99.2%–99.8% (S1 Table).

## SNP-based phylogenomic analysis of the Thai *B. glumae* strains and global strains

The genomes of the 16 Thai *B. glumae* strains were analyzed alongside publicly available genomes from multiple countries to contextualize the evolutionary relationships of Thai strains within the global population structure. A maximum-likelihood tree generated from core-genome SNP data revealed that the Thai strains were separated into multiple well-supported clades rather than clustering together (Fig 1), indicating the presence of distinct *B. glumae* populations. Interestingly, a monophyletic clade with short branch lengths and strong bootstrap support, indicating minimal SNP divergence, was

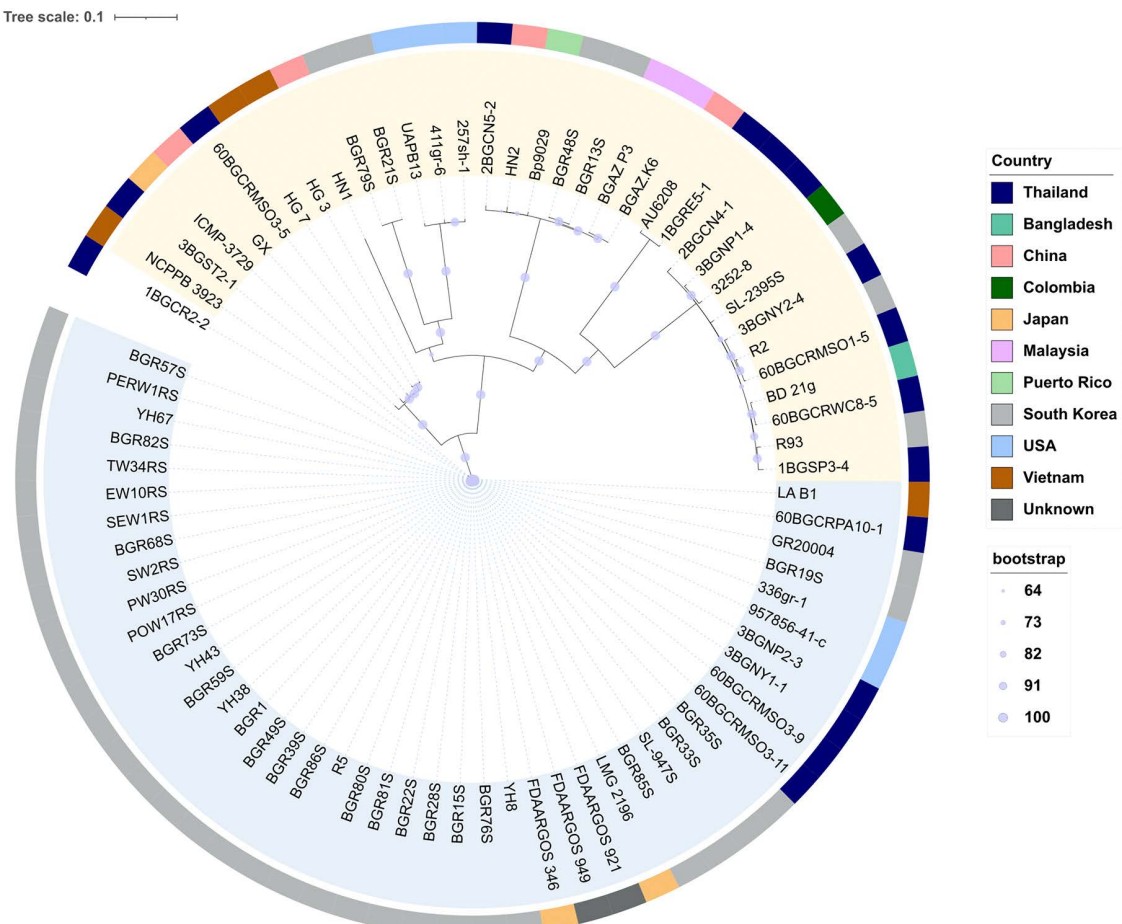

**Fig 1. Whole-genome single-nucleotide polymorphism (SNP)-based phylogenetic tree of Thai *B. glumae* strains and global reference strains.** A maximum-likelihood tree was constructed using core-genome SNPs from the Thai *B. glumae* isolates and publicly available genomes. Strain names are shown as originally recorded in public genome databases. Bootstrap support values are indicated by circles at the nodes.

observed. This clade comprised five Thai strains and strains from Korea, Japan, Vietnam, and the USA (heightened in blue, Fig 1). This suggests the presence of a clonal lineage that had spread across geographically distant regions. The other Thai strains were distributed among distinct clades with strains from various other countries (heightened in yellow Fig 1).

Among the seven strains from Chiang Rai, three strains (60BGCRPA10-1, 60BGCRMSO3-9, and 60BGCRMSO3-11) were included in the tightly clustered monophyletic clade, while the remaining Chiang Rai strains (60BGCRMSO3-5, 60BGCRMSO1-5, 60BGCRWC8-5, and 1BGCR2-2) were clustered in separate clades. Similarly, the strains from Nakhon Nayok and Nakhon Pathom were distributed across different clades. Notably, strain 1BGRE5-1 from Roi Et clustered with the human pathogenic strain AU6208.

## Distribution of plasmids and prophages in the Thai *B. glumae* strains

The presence of plasmids and prophages in the 16 Thai *B. glumae* strains was examined to gain insight into their mobi-lome composition and potential for HGT. Plasmids were identified using both reference-based (MOB-suite) and machine learning–based (PlasFlow) approaches. All strains were found to carry at least one plasmid, with several harboring multiple plasmid elements (Fig 2). Plasmid CP045089 was the most prevalent; it was detected in 81.25% (13/16) of the strains. In addition to known plasmids, several uncharacterized plasmid elements were detected. A BLAST search of the NCBI nucleotide database revealed that these unidentified elements shared partial sequence similarity with previously reported *B. glumae* plasmids. Importantly, all detected plasmids were predicted to be species specific, and no plasmids were predicted to originate from other bacterial taxa based on the tools used.

The prophage analysis revealed substantial variation across the *B. glumae* genomes, with 59 distinct prophage elements identified, comprising both complete and partial sequences (Fig 2). These prophages were linked to multiple bacterial genera, including *Burkholderia*, *Escherichia*, *Salmonella*, and *Pseudomonas*, with the majority associated with *Burkholderia*. While many of the detected prophage regions were classified as incomplete or questionable, several were predicted to be intact and potentially functional.

No clear associations were observed between the plasmid content and phylogenetic clustering of the strains, except for 60BGCRMSO3-9 and 60BGCRMSO3-11, which had a close phylogenetic relationship and identical plasmid profiles. Similarly, the prophage profiles of the isolates were highly variable, and they did not correlate with the SNP-based phylogenetic relationships. Notably, no strains exhibited identical prophage profiles.

## Putative HTGs from other *Burkholderia* species in the *B. glumae* genomes

BLAST searches of the HGT-DB led to the identification of 572 putative HTGs from other *Burkholderia* species within the *B. glumae* strains (S2 Table), with an average of 144 HTGs per genome. The results of BLAST searches of the NCBI nucleotide database confirmed the atypical presence of these genes in the *B. glumae* strains. The majority (69.1%; n = 395) of the HTGs were attributed to unclassified *Burkholderia* species. Approximately 19% (n = 109) of the HTGs were associated with plant-related *Burkholderia* species, including *Burkholderia gladioli*, *Burkholderia plantarii*, and *Burkholderia seminalis*, while 10% (n = 57) originated from members of the *B. cepacia* complex. Notably, 11 putative HTGs were from the *B. pseudomallei* group: six from *B. thailandensis*, two each from *B. pseudomallei* and *Burkholderia oklahomensis*, and one from *Burkholderia mayonis* (Fig 3, S2 Table). A functional classification of these putative HTGs revealed that 22% (n = 126) were related to phages and 15.4% (n = 88) were associated with mobile genetic elements (e.g., transposases, integrases, or recombinases) (Fig 3, S2 Table).

## Putative HTGs from *B. pseudomallei* in the *B. glumae* genomes

Two genes originally from *B. pseudomallei* that encode a putative nuclease (WP_038743760.1) and a PAAR domain-containing protein (WP_004535952.1) were detected in the 60BGCRMSO3-9 and 60BGCRMSO3-11 strains (S2 Table). Further analysis using the Alien Hunter predicted that both genes were located within a putative horizontally acquired

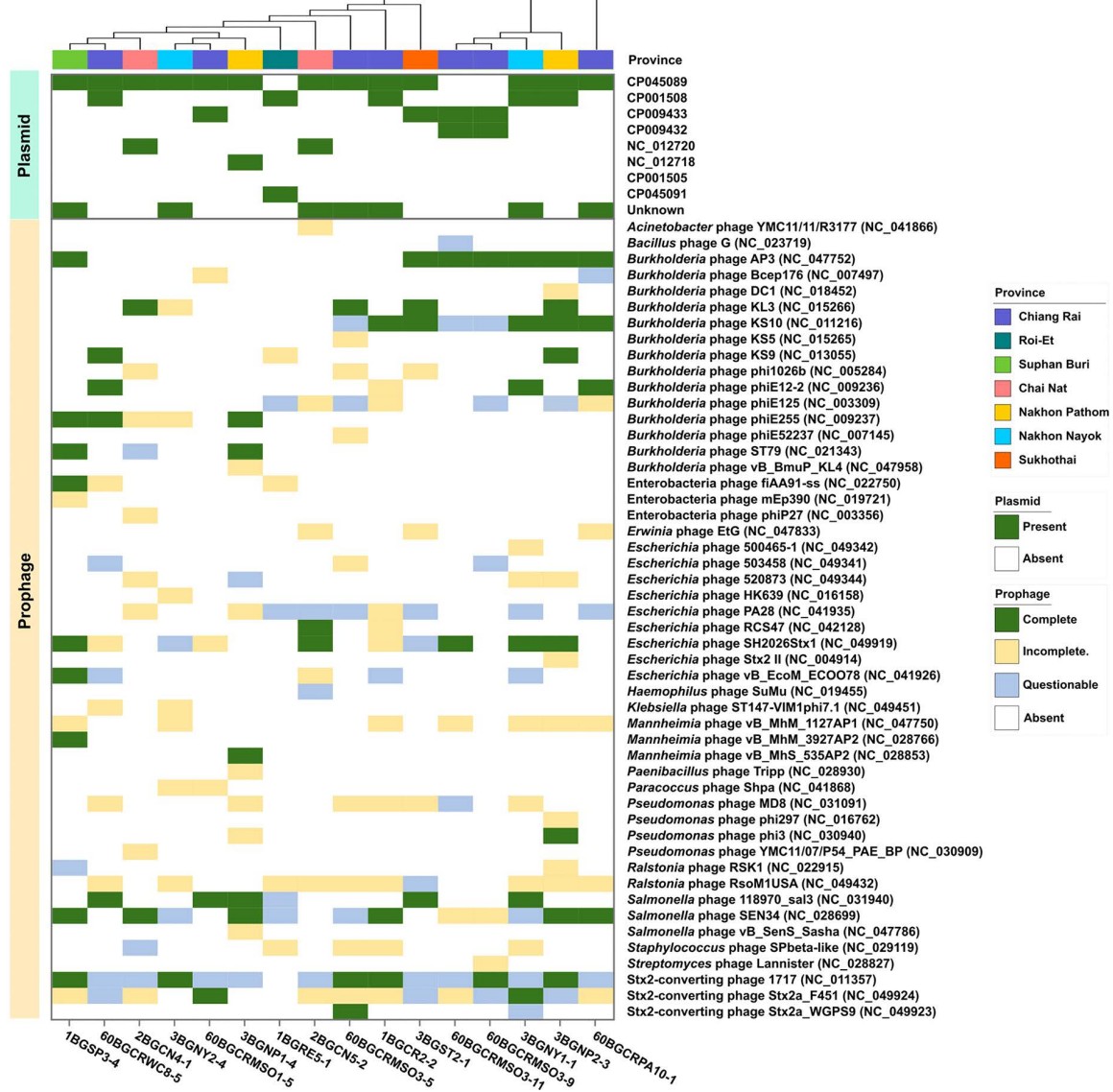

**Fig 2. Distribution of plasmids and prophages in the 16 Thai *B. glumae* genomes.** Presence–absence heatmaps of plasmids (top) and prophages (bottom) are shown. Prophages are categorized as complete (dark green), incomplete (yellow), or questionable (blue). The strains are ordered according to the core-genome SNP-based phylogeny. Each column represents an individual genome.

region (122,026–163,756) on chromosome 1 in both strains. Given that the strains were found to have the same putative HGT region and to be phylogenetically closely related, further analysis of only the 60BGCRMSO3-11 strain is described.

The putative horizontally acquired region was found to contain many hypothetical protein-coding genes (Fig 4). The remaining annotated genes in this region encode an endonuclease NucS, a helix-turn-helix domain-containing protein, phage capsid protein, YqaJ viral recombinase, DUF932 domain-containing protein, ATP-dependent nuclease, an IS3 family transposase, DNA cytosine methyltransferase, an M48 family metalloprotease, a type VI immunity family protein, a putative nuclease, a PAAR domain-containing protein, anti-phage dCTP deaminase, and tyrosine recombinase XerC (Fig 4).

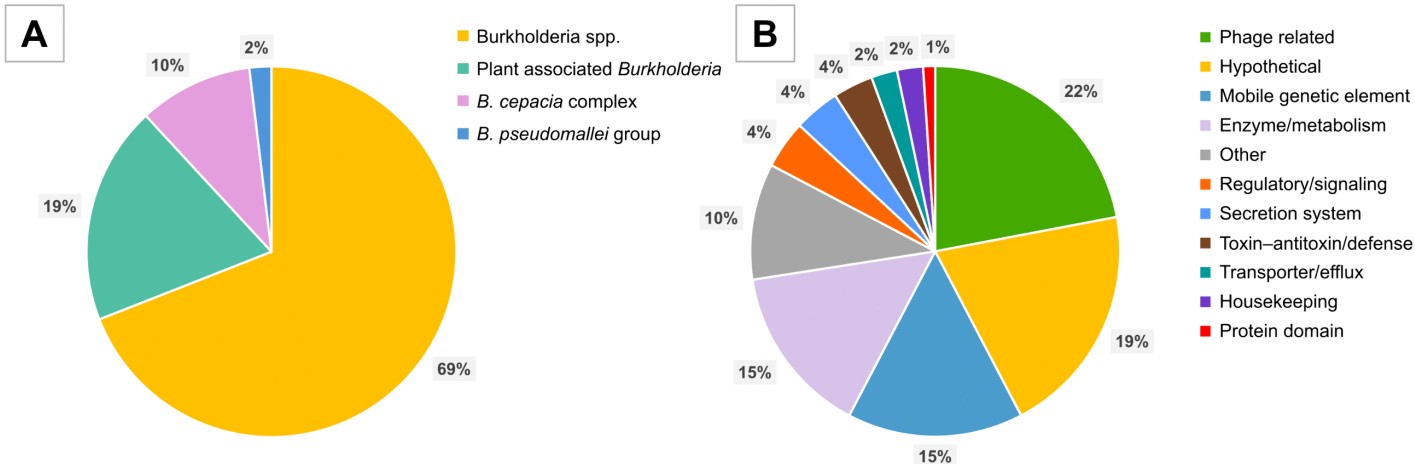

**Fig 3. Distribution of the putative horizontally transferred genes present in the *B. glumae* genomes. (A)** Distribution by species group. **(B)** Distribution by function. Percentage values are indicated within each segment. Functional categories that represent less than 1% of total HTGs are combined under "Other".

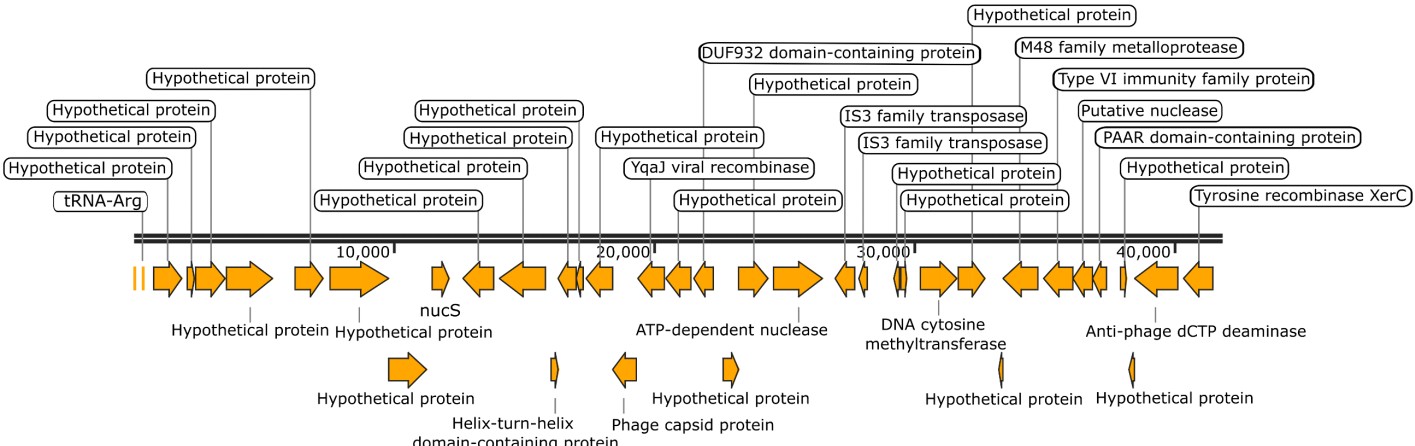

**Fig 4. Organization of genes in a putative horizontally acquired region in the *B. glumae* 60BGCRMSO3-11 strain.** The gene map of an approximately 41.7 kb chromosomal region (positions: 122,026–163,756) identified as a putative horizontal gene transfer island containing genes of *B. pseudomallei* origin is shown. Arrows indicate gene orientation.

Subsequent comparative analysis of the horizontally acquired region in the *B. glumae* 60BGCRMSO3–11 strain revealed high amino acid sequence similarity (90%–100%) with a genomic region in the *B. pseudomallei* 8400 strain (GCF_025847635.1) (Fig 5). A notable difference in the *B. glumae* 60BGCRMSO3–11 strain was the insertion of IS3 family transposase genes immediately downstream of the ATP-dependent nuclease gene (Figs 4 and 5). This resulted in a frameshift within the adjacent gene encoding a UvrD-helicase domain-containing protein (Fig 5).

The comparative analysis also showed that the horizontally acquired region shared three key genes (encoding a type VI immunity family protein, a putative nuclease, and a PAAR domain-containing protein) with *B. pseudomallei* WU_BP_O4 (GCF_041028765.1). In *B. pseudomallei* WU_BP_O4, this specific gene cassette is located between the defense island system associated with restriction–modification (DISARM) and type VI secretion system (T6SS) gene clusters.

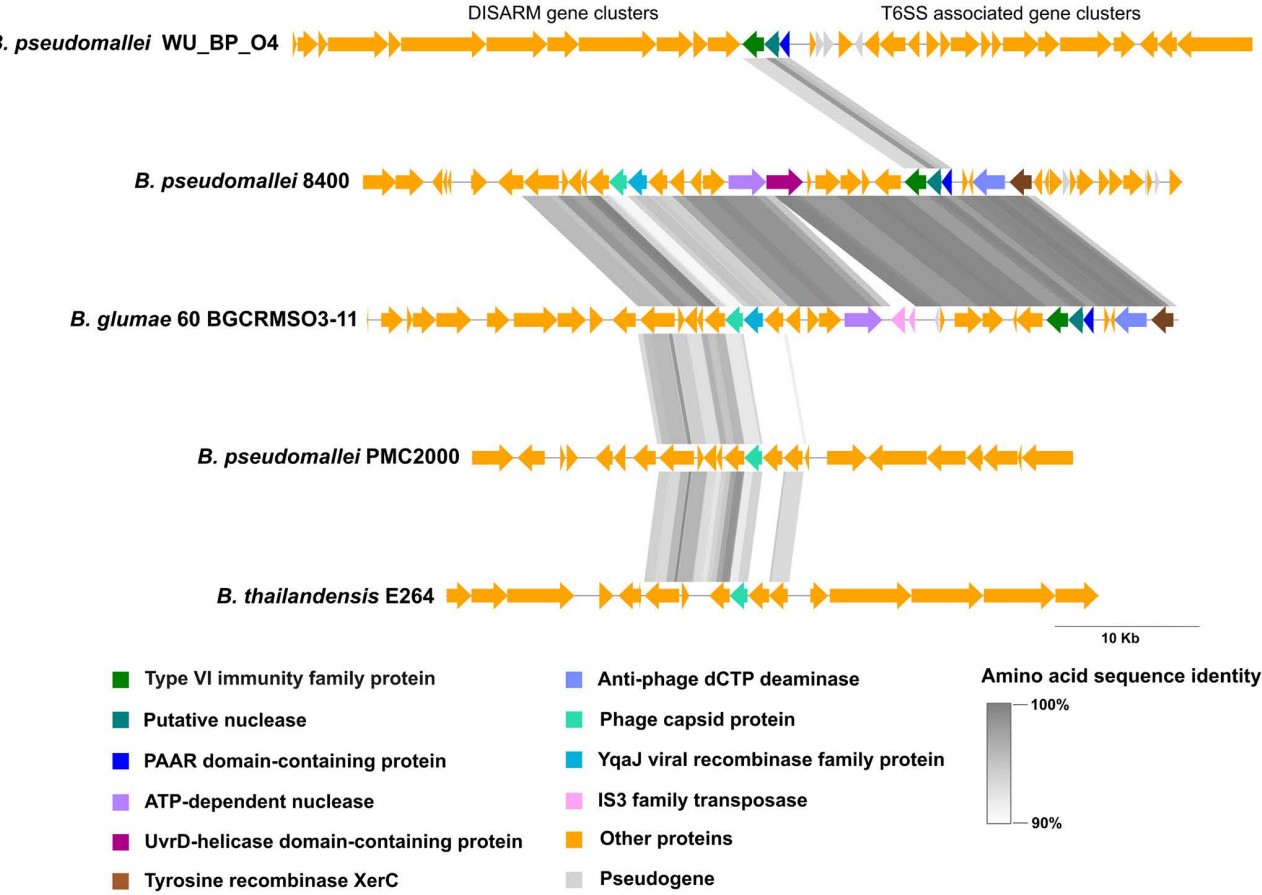

**Fig 5. Comparative analysis of the horizontally acquired region in *B. glumae* 60BGCRMSO3-11 and regions in other *Burkholderia* species.** The genomic region in the *B. glumae* 60BGCRMSO3-11 strain that spanned positions 122,026–163,756 was compared to homologous regions in *B. pseudomallei* WU_BP_O4 13, 8400, and PMC2000 and in *Burkholderia thailandensis* E264. Conserved gene blocks are indicated by shaded gray connectors, with gradient intensity representing amino acid sequence identity (90%–100%).

Furthermore, the analysis identified a cluster of hypothetical protein-coding genes and phage capsid protein-coding genes in the horizontally acquired region that were highly similar to those found in other members of the *B. pseudomallei* complex, namely *B. pseudomallei* PMC2000 (CP025302.1) and *B. thailandensis* E264 (CP000086.1) (Fig 5).

## Discussion

The findings of this study provide new insights into the genomic composition of *B. glumae* populations in Thailand and support the hypothesis that HGT plays an important role in shaping the genomic plasticity of this pathogen. Previous genomic studies of *B. glumae* have largely focused on pathogenicity factors, host interactions, and population diversity, with limited investigation into HGT. Although previous studies have reported HGT events from distinct soil bacterial taxa such as *Pseudomonas* and *Streptomyces* [14,15], the investigation of HGT between *B. glumae* and other *Burkholderia* species remains limited, particularly in environments where *B. glumae* shares the same habitat with human-pathogenic *Burkholderia*, as observed in Thailand.

The SNP-based phylogenomic analysis, which included global isolates, revealed considerable genetic diversity among the 16 Thai *B. glumae* strains. Because *B. glumae* in Thailand is thought to be a recently emerging and imported

pathogen [4], limited diversity would be expected. The observed genomic heterogeneity among the Thai strains therefore supports a polyclonal population structure rather than expansion from a single introduction. In addition, the identification of a distinct monophyletic clade comprising five Thai strains and database-retrieved strains from Korea, Japan, Vietnam, and the USA suggests the presence of a globally disseminated clonal lineage. Conversely, the distribution of the other Thai strains across different clades, even those from the same geographic region, indicates the co-circulation of multiple genetically distinct lineages, possibly due to multiple introductions. These findings align with those of prior studies that reported the presence of a widespread monophyletic clade and the co-occurrence of phylogenetically distinct *B. glumae* strains within the same region [34,35].

In addition to core-genome diversity, the 16 *B. glumae* strains exhibited substantial variability in their plasmid and prophage elements. All 16 Thai *B. glumae* strains were predicted to contain at least one plasmid, which is consistent with the findings of previous genomic studies showing that plasmids are commonly found in *B. glumae* [34,35]. Notably, all predicted plasmids were specific to *B. glumae*, and no plasmids were predicted to be derived from other bacterial taxa, indicating species-specific plasmid content among the Thai isolates.

A high degree of prophage heterogeneity was observed, with no strains exhibiting identical prophage profiles. Additionally, the prophage profiles did not correlate with phylogenetic clustering or isolation regions, suggesting the occurrence of independent acquisition events. While most of the prophage regions were classified as incomplete or questionable, several were predicted to be intact and potentially functional, particularly those with similarities to known *Burkholderia* phage sequences. This suggests that the studied *B. glumae* strains may have the capacity to generate phages, which could contribute to their genomic plasticity and increase the frequency of HGT [36]. Others have also reported that *Burkholderia* species, including *B. glumae* and *B. pseudomallei*, have diverse prophage repertoires, reflecting their capacity to serve as reservoirs for mobile elements and gene transfer vectors [16,37–40].

Previous genomic studies have shown that *Burkholderia* species frequently acquire DNA from both intra- and inter-genus sources and that the process is often mediated by integrative elements, transposons, or phages [12–15,38]. In this study, we identified several putative HTGs in the genomes of the Thai *B. glumae* strains that were potentially acquired from other *Burkholderia* species. While most of these genes appeared to originate in uncharacterized or plant-associated *Burkholderia* species, genes from human-pathogenic *Burkholderia* species were also detected. Notably, most of the putative HTGs were phage-related genes, which suggests that HGT involving *B. glumae* was primarily mediated by phages.

An important finding of this study was the identification of putative HTG regions in two closely related strains (60BGCRMSO3-9 and 60BGCRMSO3-11). They were found to share a chromosomal island carrying genes very similar to those found in *B. pseudomallei* 8400. Several genes within this region were found to be associated with mobile genetic elements and phage defense mechanisms. The presence of genes coding for IS3 family transposase, tyrosine recombinase XerC, and YqaJ viral recombinase suggests that transposition and recombination events may have facilitated gene transfer [41–43]. In addition, the presence of ATP-dependent nuclease, UvrD-helicase domain-containing protein, and antiphage dCTP deaminase genes implies the acquisition of DNA repair, foreign DNA degradation, and phage resistance mechanisms [44–46]. Notably, a phage capsid gene was also present in this region; however, the absence of other essential phage genes suggests that it is unlikely that the region encodes functional prophages. Instead, this gene may contribute to antiphage defense [47]. A cluster of genes encoding a T6SS immunity family protein, a putative nuclease, and a PAAR domain-containing protein was also detected. These genes are likely associated with T6SS, with the nuclease potentially functioning as an effector [48–50]. The PAAR domain is known to sharpen the puncturing tip of the T6SS and is often associated with the specific delivery of toxic effectors, such as nucleases, into competing bacterial cells [49]. The presence of an adjacent T6SS immunity family protein gene is further evidence that this gene cluster is associated with T6SS [50]. In addition, this gene cluster was also identified in *B. pseudomallei* WU_BP_O4, and its positioning near this strain's T6SS gene clusters indicates an association with T6SS. Furthermore, *B. glumae* is known to possess multiple distinct T6SS gene clusters involved in virulence and interspecies interactions [51]. Therefore, the acquisition of

these T6SS-related components may provide *B. glumae* with enhanced virulence and a competitive advantage over other microorganisms in the environment. In addition, the cluster of genes encoding phage capsid and hypothetical proteins within the putative HGT region of the two *B. glumae* strains was similar to those found in members of the *B. pseudomallei* complex, including *B. thailandensis*. Given that this cluster was integrated at different genomic loci in the strains, it is likely to represent a mobile genetic element. As the majority of genes within this cluster encode hypothetical proteins, further research is required to elucidate their roles in HGT and to determine their potential biological functions. Collectively, these results suggest that the acquisition of *B. pseudomallei* genes via HGT enhances the genomic plasticity of *B. glumae* and may contribute to its ability to compete with other microbes in the environment. The presence of phage-related genes in the horizontally transferred regions implies that the HGT event(s) may have been mediated by phage or antiphage mechanisms [47].

Studies have shown that co-localization in shared environmental niches promotes gene exchange between environmental and pathogenic bacteria [52,53]. While *B. pseudomallei* is highly prevalent in northeast Thailand [5], it has also been detected in northern Thailand, including Chiang Rai province [54], where the *B. glumae* strains with putative HTG regions were isolated. This geographic co-occurrence provides direct ecological opportunity for interspecies contact. In addition, both species exhibit higher prevalence during periods of increased rainfall and humidity [2,6], which lead to simultaneous population peaks during the rice-growing season. These seasonal conditions create dense, metabolically active microbial communities in waterlogged paddy soils, enhancing the frequency of cell-to-cell contact, and the mobilization of plasmids and prophages [7]. Phage activity is also likely to be intensified under these environmentally stressed, nutrient-fluctuating conditions. Phages are known to adapt to such environments by modulating infection strategies, expanding host ranges, and expressing auxiliary metabolic genes that enhance microbial stress tolerance [8]. Moreover, phage-driven processes such as phage predation and the viral shunt can increase bacterial DNA release and uptake, activate mobile genetic elements, and thereby enhance the potential for phage-mediated horizontal gene transfer [8,55]. In addition, bacterial antiviral systems, which balance the costs of cell lysis with the benefits of acquiring new genes, may further modulate the likelihood of successful gene exchange [8]. Consequently, in rice-field settings where *B. glumae* and *B. pseudomallei* overlap, these combined ecological and mechanistic drivers likely facilitate interspecies HGT, offering a plausible explanation for the gene transfer events observed in this study.

The transfer of genes from *B. pseudomallei* to *B. glumae* has potentially significant clinical implications. A previous report of *B. glumae* isolated from an immunocompromised patient [3] demonstrates that the species has the capacity to cause human infection. Moreover, the transfer of virulence genes from *B. pseudomallei* to other *Burkholderia* species has also been reported; for example, *B. thailandensis* and *B. cepacia* complex members have been shown to acquire capsular biosynthesis loci that resemble those of *B. pseudomallei* [11–13]. Our findings suggest that similar gene flow may be occurring between *B. pseudomallei* and *B. glumae*. The acquisition of virulence-associated genes by *B. glumae* from human-pathogenic *Burkholderia* could have important consequences, potentially enhancing its pathogenicity, expanding its host range, and altering its ecological behavior. These observations underscore the need for continued genomic surveillance using One Health perspective to monitor the emergence of traits that may impact both agricultural productivity and public health.

Several limitations should be acknowledged. Although the 16 isolates analyzed in this study were collected from multiple geographic regions and different years, the overall sample size remains small, which may limit the generalizability of the observed population structure and HGT patterns. In addition, our conclusions are based solely on genomic data, without functional validation of the putative horizontally transferred genes. Future studies should focus on the characterization of the genes transferred from *B. pseudomallei* to *B. glumae* to determine their expression, biological functions, and roles in bacterial interactions and pathogenesis. In addition, expanding the sample size and incorporating temporal and ecological data through systematic surveillance would provide deeper insights into the frequency and patterns of HGT events.

## Conclusions

Our findings indicate extensive genomic heterogeneity among *B. glumae* strains in Thailand and provide evidence of horizontal gene acquisition from *B. pseudomallei.* The detection of *B. pseudomallei*-derived genes within *B. glumae* chromosomes underscores the potential for cross-species genetic exchange in shared environments. These findings underscore the importance of environmental surveillance and genome-based monitoring to track the emergence of novel genomic combinations that may impact plant and/or human health.

## Supporting information

**S1 Table. Assembly quality control and taxonomic analysis.**
(XLSX)

**S2 Table. Putative horizontally transferred genes.**
(XLSX)

## Acknowledgments

We thank Dr. Kristen Sadler from Scribendi (www.scribendi.com) for editing a draft of this manuscript.

## Author contributions

**Conceptualization:** Sujin Patarapuwadol, Woranich Hintong, Jiraphan Premsuriya.

**Data curation:** Sujin Patarapuwadol, Woranich Hintong, Jiraphan Premsuriya.

**Formal analysis:** Pornpavee Nualnisachol, Natnicha Wankaew, Worarat Kruasuwan, Thanchanok Sawaengwong, Phatcharin Laosena, Jiraphan Premsuriya.

**Funding acquisition:** Jiraphan Premsuriya.

**Investigation:** Sujin Patarapuwadol, Pornpavee Nualnisachol, Jiraphan Premsuriya.

**Methodology:** Pornpavee Nualnisachol, Natnicha Wankaew, Worarat Kruasuwan, Thanchanok Sawaengwong, Phatcharin Laosena, Jiraphan Premsuriya.

**Project administration:** Phatcharin Laosena, Jiraphan Premsuriya.

**Resources:** Sujin Patarapuwadol.

**Software:** Pornpavee Nualnisachol.

**Supervision:** Sujin Patarapuwadol, Woranich Hintong.

**Validation:** Sujin Patarapuwadol, Pornpavee Nualnisachol.

**Visualization:** Pornpavee Nualnisachol, Jiraphan Premsuriya.

**Writing – original draft:** Sujin Patarapuwadol, Jiraphan Premsuriya.

**Writing – review & editing:** Sujin Patarapuwadol, Woranich Hintong, Pornpavee Nualnisachol, Natnicha Wankaew, Worarat Kruasuwan, Thanchanok Sawaengwong, Phatcharin Laosena, Jiraphan Premsuriya.

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
