## [Decision Letter · Decision Letter 0]

10 Nov 2025

Dear Dr. Premsuriya,

https://journals.plos.org/plosone/s/submission-guidelines#loc-laboratory-protocols . Additionally, PLOS ONE offers an option for publishing peer-reviewed Lab Protocol articles, which describe protocols hosted on protocols.io. Read more information on sharing protocols at https://plos.org/protocols?utm_medium=editorial-email&utm_source=authorletters&utm_campaign=protocols .

We look forward to receiving your revised manuscript.

Kind regards,

Annamaria Bevivino

Academic Editor

PLOS ONE

**Journal Requirements:**

“This research was supported by Chulabhorn Royal Academy (https://www.cra.ac.th/). Fundamental Fund: fiscal year 2024 by National Science Research and Innovation Fund, grant number 198501.”

4. Please note that funding information should not appear in any section or other areas of your manuscript. We will only publish funding information present in the Funding Statement section of the online submission form. Please remove any funding-related text from the manuscript.

5. Please note that your Data Availability Statement is currently a direct link to access each database. If your manuscript is accepted for publication, you will be asked to provide these details on a very short timeline. We therefore suggest that you provide this information now, though we will not hold up the peer review process if you are unable.

**Additional Editor Comments:**

The manuscript has been revised by two experts in the field who provided constructive suggestion to further improve the clarity and quality of the work.  Please, address each comment carefully and submit a revised version along with a point-by-point response to the reviewers' comments.  

The study provides valuable insights into the genomic heterogeneity and interspecies gene transfer between* B. glumae*  and* B. pseudomallei. * Below, additional suggestion:

In the Introduction, please clarify the ecological overlap between *B. glumae*  and* B. pseudomalleii, * briefly stating how this shared niche could facilitate genetic exchange.

Highlight novelty and research gap, by explicitlying state what remains unknown (e.g., genomic evidence of interspecies HGT in B. glumae from Thailand).Emphasize rationale and objectives, explaining why whole-genome sequencing and comparative genomics are the best approaches for addressing the study question.Strengthen the opening paragraph of the Discussion, expanding slightly on how your findings relate to existing knowledge on *B. glumae * genomics and HGT in *Burkholderia * species.Discuss possible mechanisms or ecological contexts facilitating HGT (e.g., soil or plant-associated microbial communities in rice fields).Mention any study constraints such as small sample size, limited geographic or temporal coverage, or lack of expression data for transferred genes.Revise the final paragraph of the Discussion to strenghten the conclusion and emphasize the broader implication of the study for both plant and human health.

Reviewers' comments:

Reviewer's Responses to Questions

**Comments to the Author**

1. Is the manuscript technically sound, and do the data support the conclusions?

Reviewer #1: Yes

Reviewer #2: Yes

2. Has the statistical analysis been performed appropriately and rigorously?

Reviewer #1: Yes

Reviewer #2: N/A

3. Have the authors made all data underlying the findings in their manuscript fully available?

Reviewer #1: Yes

Reviewer #2: Yes

4. Is the manuscript presented in an intelligible fashion and written in standard English?

Reviewer #1: Yes

Reviewer #2: Yes

Reviewer #1: The manuscript aims to investigate the potential for gene transfer between Burkholderia glumae and Burkholderia pseudomallei. Methods such as whole-genome sequencing (WGS) and comparative genomic analysis are likely to be used to determine whether genetic material was exchanged. In this study, 16 B. glumae strains, isolated from rice fields in 7 Thai provinces between 2011 and 2017, underwent WGS, and 36 previously published B. glumae genomes available in GenBank were also used. The results showed high genetic diversity and a polyclonal population structure. A clonal lineage was detected spreading across geographically distant regions. Plasmids, prophage elements, and several putative horizontally transferred genes were detected in all isolates. Two strains were found to share a chromosomal island that carries genes that are highly similar to those found in B. pseudomallei. This provides evidence of horizontal gene transfer between species.

General comment

Overall, the submitted manuscript is well-written and scientifically sound. I did not identify any scientific errors in the hypothesis or experimental design. Some issues need to be addressed to improve the quality and readability of the work.

Issues to be addressed:

As stated in the text, strains 60BGCRMSO1-5, 60BGCRMSO3-5, 60BGCRMSO3-9, 60BGCRMSO3-11, 60BGCRWC8-5, and 60BGCRPA10-1 (reported in Table 1) were previously subjected to whole genome sequencing (WGS) in another manuscript (https://doi.org/10.3390/pathogens11060676). Therefore, I suggest distinguishing between re-sequenced and de novo sequenced strains in Table 1. Were the results obtained by re-sequencing consistent with those previously obtained? Furthermore, I recommend adding columns for the geographic coordinates of the isolation locations and the rice cultivars to Table 1.

L96. Why were B. glumae isolates collected from rice fields grown at 37°C rather than at a lower temperature for DNA extraction?

Fig.2 should be modified to improve clarity. The province at the top of the heatmap is not very clear.

Please use the same abbreviation for Average Nucleotide Identity (ANI) in the text and in Table S1. I suggest writing the full name of ANI in the legend of the table.

L346. B. pseudomallei should be written in italics.

Reviewer #2: General Comment

The authors present an exploratory comparative genomics study on several Burkholderia strains isolated from Thailand, using various bioinformatic tools to identify potential mobile genetic elements and predict horizontal gene transfer (HGT) events via conjugation and transduction. The work is primarily hypothesis-generating, providing computational predictions that could inform future experimental validation.

Overall, the data could be of value to the field. However, several aspects of the manuscript would benefit from clarification and refinement to improve scientific precision and presentation.

Major and Minor Comments

Line 26 – The term genomic plasticity is not sufficiently defined. Please clarify what is meant in this context (e.g., genome rearrangements, mobile elements, recombination, or overall genomic variability).

Line 83 – If the putative mobile genetic elements were transferred in the past, the past tense should be used.

Line 99 – The genomic DNA extraction method (scraping colonies from agar plates) is suboptimal, often resulting in lower-quality DNA. While this may have sufficed for sequencing, it is not ideal. Please provide justification or references supporting this method.

Line 147 – Consider citing relevant literature describing the specific Burkholderia strains mentioned, to provide more context and support for the statements made.

Line 162 – The statement that “the strains were distributed across clades” is somewhat tautological; any phylogenetic analysis will, by definition, place strains across clades. Consider rephrasing to highlight a meaningful observation (e.g., clustering pattern, host-specific associations, or geographic structure).

Line 164 – The statement that strains had minimal SNP divergence seems inconsistent with the earlier claim of high genetic diversity. Please clarify or reconcile these points.

Line 166 – Past tense (“had spread”) would be more appropriate if referring to a historical dissemination event.

Line 193 – Please clarify that this represents a negative finding (absence of prediction) rather than evidence for the absence of the feature, or provide data supporting the conclusion.

Line 282 – Use the past tense (“had occurred”) for events of horizontal gene transfer. Moreover, because genomic plasticity is not clearly defined, it is difficult to interpret the meaning of this statement.

Line 284 – The phrasing about high genetic diversity could be reconsidered. High genomic diversity is expected among environmental Burkholderia isolates, so please specify in what sense this observation is noteworthy.

Line 288 – Please clarify whether the strains were isolated in Thailand or processed there by a research group; the current wording is ambiguous.

Line 296 – The plasmids are only predicted to exist; please make this clear in the text.

Line 308 – The phrase “facilitate HGT” suggests a mechanistic enhancement, which is not demonstrated here. “Increase the frequency of HGT” may be more accurate.

Line 319 – This event appears to have occurred in the past; please adjust verb tense accordingly.

Line 366 – The use of “may” is confusing in this context. If the strain is a clinical isolate, it must have caused infection at some point. Consider rephrasing to something like: “The strain may further increase pathogenicity” or “The isolate shows genomic traits consistent with pathogenic potential.”

Figures

Figure 1 – The rationale for including all listed strains is unclear, and the naming convention is nonstandard. Please explain the context, significance, and nomenclature choice.

Figure 2 – The figure titles are rotated and difficult to read. Please reformat for clarity and ensure proper orientation.

Figure 3 – The figure’s implications are unclear. Colors are difficult to distinguish, and no numerical percentages are provided. The category “Other” further complicates interpretation. Please explain the biological relevance of the data shown — why the relative percentages matter and how they support your conclusions.

**Do you want your identity to be public for this peer review?** For information about this choice, including consent withdrawal, please see our Privacy Policy

Reviewer #1: **Yes: ** Silvia Tabacchioni

Reviewer #2: No

---

## [Author Response · Author response to Decision Letter 1]

3 Dec 2025

Response to the reviewers’ comments

Editor Comments:

The study provides valuable insights into the genomic heterogeneity and interspecies gene transfer between B. glumae and B. pseudomallei. Below, additional suggestion:

In the Introduction, please clarify the ecological overlap between B. glumae and B. pseudomalleii, briefly stating how this shared niche could facilitate genetic exchange.

We thank the editor for this helpful suggestion. We have revised the Introduction to more clearly describe the ecological overlap between B. glumae and B. pseudomallei, and to explain how this shared habitat may promote opportunities for genetic exchange.

The original text:

“Given that B. glumae and B. pseudomallei share the environmental niche of rice fields, it is highly possible that they coexist in Thai rice fields and interact directly. However, few studies have investigated the specific interactions between B. glumae and B. pseudomallei.”

Has been revised to:

“Given that B. glumae and B. pseudomallei share the environmental niche of rice fields, it is highly likely that they coexist in Thai rice paddies and interact directly. In addition, the prevalence of both species increases with higher humidity and rainfall [2,6]. As a result, the population densities of both organisms are likely to peak simultaneously during the high-precipitation rice-growing season. The coincident population dynamics of B. glumae and B. pseudomallei may intensify their niche sharing, resulting in frequent cell-to-cell contact within the highly active and dense microbial communities of waterlogged rice-field soils. These high-density and environmentally stressed conditions provide an optimal setting for genetic exchange, thereby increasing the likelihood of horizontal gene transfer (HGT) mediated by mobile genetic elements or bacteriophages [7,8].”

Highlight novelty and research gap, by explicitly state what remains unknown (e.g., genomic evidence of interspecies HGT in B. glumae from Thailand).

We thank the editor for this constructive feedback. We have revised the Introduction to more clearly emphasize the novelty of our study and to explicitly identify the existing research gap. In particular, we now state that genomic evidence of interspecies HGT in B. glumae isolates from Thailand has not been previously investigated.

The original text:

“Together with the knowledge that B. glumae and B. pseudomallei share an environmental habitat in Thailand, the documented capacity for HGT within the Burkholderia genus indicates that it is critical to determine whether these species exchange genes. Such gene exchange may have implications for the evolution of virulence in B. glumae and may alter the pathogenic landscape in agricultural and public health contexts.”

Has been revised to:

“The coexistence of B. glumae and B. pseudomallei in Thai environments, coupled with extensive evidence of horizontal gene transfer within Burkholderia, makes it critical to determine whether these two species exchange genes in nature. However, no study to date has examined genomic evidence of interspecies HGT in B. glumae isolates from Thailand. Addressing this gap is essential, as such gene exchange may influence the evolution of virulence, environmental adaptation, and the broader pathogenic landscape with implications for both agriculture and public health.”

Emphasize rationale and objectives, explaining why whole-genome sequencing and comparative genomics are the best approaches for addressing the study question.

Thank you for this very helpful suggestion. We have revised the Introduction to clearly describe the rationale for using whole-genome sequencing (WGS) and comparative genomics, and to explain why these methods are appropriate for addressing our study objectives.

The original text:

“In this study, we performed whole-genome sequencing (WGS) and a comparative genomic analysis of 16 B. glumae isolates from Thailand. Our objective was to evaluate the potential for HGT events between B. glumae and other Burkholderia species, particularly B. pseudomallei. Our findings provide new insights into the evolutionary dynamics of B. glumae and contribute to a better understanding of its genomic relationships with clinically important Burkholderia species.”

Has been revised to:

“In this study, we performed whole-genome sequencing (WGS) and a comparative genomic analysis of 16 B. glumae isolates from Thailand to investigate evidence of HGT events between B. glumae and other Burkholderia species, particularly B. pseudomallei. WGS combined with comparative genomics allows high-resolution detection of mobile genetic elements, horizontally acquired regions, and evolutionary relationships across species. This approach is therefore well suited for identifying HGT events and elucidating genome evolution in Burkholderia [14,15]. Our findings provide new insights into the evolutionary dynamics of B. glumae and contribute to a better understanding of its genomic relationships with clinically important Burkholderia species.”

Strengthen the opening paragraph of the Discussion, expanding slightly on how your findings relate to existing knowledge on B. glumae genomics and HGT in Burkholderia species.

We thank the editor for this valuable comment. We have revised the opening paragraph of the Discussion to better contextualize our findings within the existing literature on B. glumae genomics and HGT in Burkholderia species. The revised text now more clearly highlights what is known, what remains insufficiently studied, and how our findings advance current understanding.

The original text:

“The findings of this study provide insights into the genomic composition of B. glumae populations in Thailand and support the hypothesis that HGT plays an important role in shaping the genomic plasticity of this pathogen. Furthermore, they indicate that interspecies HGT may occur between B. glumae and B. pseudomallei.”

Has been revised to:

“The findings of this study provide new insights into the genomic composition of B. glumae populations in Thailand and support the hypothesis that HGT plays an important role in shaping the genomic plasticity of this pathogen. Previous genomic studies of B. glumae have largely focused on pathogenicity factors, host interactions, and population diversity, with limited investigation into HGT. Although previous studies have reported HGT events from distinct soil bacterial taxa such as Pseudomonas and Streptomyces [14,15], the investigation of HGT between B. glumae and other Burkholderia species remains limited, particularly in environments where B. glumae shares the same habitat with human-pathogenic Burkholderia, as observed in Thailand.”

Discuss possible mechanisms or ecological contexts facilitating HGT (e.g., soil or plant-associated microbial communities in rice fields).

Thank you for this insightful comment. In response, we have substantially expanded the Discussion to describe the ecological and mechanistic factors that may facilitate horizontal gene transfer (HGT) between B. glumae and B. pseudomallei in rice-field environments.

The original text:

“Thus, in areas of Thailand where rice fields harbor both B. glumae and B. pseudomallei, horizontal transfer of genes between these species is possible. Such HGT may be driven by selective pressures, such as environmental stress, phage predation, and microbial competition [55].”

Has been revised to:

“This geographic co-occurrence provides direct ecological opportunity for interspecies contact. In addition, both species exhibit higher prevalence during periods of increased rainfall and humidity [2,6], which lead to simultaneous population peaks during the rice-growing season. These seasonal conditions create dense, metabolically active microbial communities in waterlogged paddy soils, enhancing the frequency of cell-to-cell contact, and the mobilization of plasmids and prophages [7]. Phage activity is also likely to be intensified under these environmentally stressed, nutrient-fluctuating conditions. Phages are known to adapt to such environments by modulating infection strategies, expanding host ranges, and expressing auxiliary metabolic genes that enhance microbial stress tolerance [8]. Moreover, phage-driven processes such as phage predation and the viral shunt can increase bacterial DNA release and uptake, activate mobile genetic elements, and thereby enhance the potential for phage-mediated horizontal gene transfer [8,55]. In addition, bacterial antiviral systems, which balance the costs of cell lysis with the benefits of acquiring new genes, may further modulate the likelihood of successful gene exchange [8]. Consequently, in rice-field settings where B. glumae and B. pseudomallei overlap, these combined ecological and mechanistic drivers likely facilitate interspecies HGT, offering a plausible explanation for the gene transfer events observed in this study.”

Mention any study constraints such as small sample size, limited geographic or temporal coverage, or lack of expression data for transferred genes.

Thank you for this helpful suggestion. We have added a limitations statement prior to the section on future directions in the Discussion. The added text reads:

“Several limitations should be acknowledged. Although the 16 isolates analyzed in this study were collected from multiple geographic regions and different years, the overall sample size remains small, which may limit the generalizability of the observed population structure and HGT patterns. In addition, our conclusions are based solely on genomic data, without functional validation of the putative horizontally transferred genes.”

Revise the final paragraph of the Discussion to strengthen the conclusion and emphasize the broader implication of the study for both plant and human health.

Thank you for this valuable suggestion. We have revised the final paragraph of the Discussion to strengthen the overall conclusion and emphasize the broader implications of our findings for both plant and human health.

The original text:

“Thus, the acquisition of virulence genes by B. glumae from human pathogenic Burkholderia species, particularly B. pseudomallei, may increase B. glumae’s pathogenicity and host range.”

Has been revised to:

“Our findings suggest that similar gene flow may be occurring between B. pseudomallei and B. glumae. The acquisition of virulence-associated genes by B. glumae from human-pathogenic Burkholderia could have important consequences, potentially enhancing its pathogenicity, expanding its host range, and altering its ecological behavior. These observations underscore the need for continued genomic surveillance using One Health perspective to monitor the emergence of traits that may impact both agricultural productivity and public health.”

Review Comments to the Author

Reviewer 1:

As stated in the text, strains 60BGCRMSO1-5, 60BGCRMSO3-5, 60BGCRMSO3-9, 60BGCRMSO3-11, 60BGCRWC8-5, and 60BGCRPA10-1 (reported in Table 1) were previously subjected to whole genome sequencing (WGS) in another manuscript (https://doi.org/10.3390/pathogens11060676). Therefore, I suggest distinguishing between re-sequenced and de novo sequenced strains in Table 1. Were the results obtained by re-sequencing consistent with those previously obtained? Furthermore, I recommend adding columns for the geographic coordinates of the isolation locations and the rice cultivars to Table 1.

Thank you for this helpful comment. We have updated Table 1 to clearly distinguish between de novo–sequenced strains and strains that were previously sequenced and re-sequenced in this study. An additional column titled “Sequencing status” has been added to indicate whether each isolate underwent de novo sequencing or re-sequencing. We also added a column for rice cultivar, as suggested. However, geographic coordinates were not available for these isolates; therefore, we retained the province of origin, which represents the most precise location information available for this dataset.

To reflect these changes, the Table 1 title has also been updated to:

“Table 1. Isolation information and sequencing status of B. glumae strains analyzed in this study.”

For the strains that were re-sequenced, the assemblies generated in our study provide substantially better completeness compared with the assemblies in the previous report (https://doi.org/10.3390/pathogens11060676). The previous study used only short-read sequencing, which resulted in highly fragmented assemblies (>100 contigs) and limited the downstream analyses such as determining gene localization, reconstructing plasmids, and mapping transposable elements. In this study, we employed a hybrid sequencing approach that combined short- and long-read data to obtain complete genome assemblies, including closed chromosomes and plasmids. These high-quality assemblies made it possible to accurately assess genomic localization, plasmid composition, and transposable elements.

L96. Why were B. glumae isolates collected from rice fields grown at 37°C rather than at a lower temperature for DNA extraction?

Thank you for pointing out this issue. The temperature reported in the manuscript was incorrect. The isolates were actually grown at 35°C, not 37°C. We have corrected this in the revised manuscript. The use of 35°C follows the optimum temperature for B. glumae from previous report (30-35°C) (doi: 10.1111/j.1364-3703.2010.00676.x). No steps in our workflow involved incubation at 37°C.

We selected 35°C over 30°C to optimize DNA purity for Nanopore sequencing. DNA extracted from cultures grown at 30°C consistently showed low A260/230 ratios, likely due to metabolite contamination that was not efficiently removed by the extraction kit. Such metabolites can adversely affect long-read sequencing performance. Cultivating the isolates at 35°C yielded DNA of substantially higher purity suitable for high-quality Nanopore sequencing.

Fig.2 should be modified to improve clarity. The province at the top of the heatmap is not very clear.

Thank you for this helpful comment regarding the clarity of Figure 2. To address this, we have updated the figure by modifying the color palette used for province annotations to enhance visual distinction and readability. In addition, we increased the size of the annotation boxes to ensure that the labels are more easily interpretable. The revised Figure 2 now provides clearer visualization of province-level metadata.

Please use the same abbreviation for Average Nucleotide Identity (ANI) in the text and in Table S1. I suggest writing the full name of ANI in the legend of the table.

We appreciate the reviewer’s careful review of the supplementary table and for pointing out this correction. In response, we have corrected the typographical error in Table S1 by replacing AIN with ANI, and we have added the full term “Average Nucleotide Identity (ANI)” to the legend of Table S1 for clarity.

L346. B. pseudomallei should be written in italics.

We appreciate the reviewer for pointing out this formatting detail. The genus–species name B. pseudomallei has now been corrected to italics at Line 346.

Reviewer 2:

Line 26 – The term genomic plasticity is not sufficiently defined. Please clarify what is meant in this context (e.g., genome rearrangements, mobile elements, recombination, or overall genomic variability).

We thank the reviewer for this helpful suggestion. The sentence has been revised to clarify what is meant by “genomic plasticity.” Specifically, we now define genomic plasticity as including frequent genome rearrangements, variability in mobile genetic elements, and recombination events that facilitate horizontal gene transfer.

The original text:

“Given the high genomic plasticity and frequent horizontal gene transfer observed in Burkholderia species, there are concerns about the emergence of novel traits that may affect both plant and human health.”

Has been revised to:

“Given the high genomic plasticity of Burkholderia species, including frequent genome rearrangements, variability in mobile genetic elements, and recombination events that facilitate horizontal gene transfer, there are concerns about the emergen

---

## [Editor Report · Decision Letter 1]

16 Dec 2025

Whole-genome sequencing of Burkholderia glumae strains from Thailand reveals potential horizontal gene transfer with Burkholderia pseudomallei

PONE-D-25-54603R1

Dear Dr. Premsuriya,

We’re pleased to inform you that your manuscript has been judged scientifically suitable for publication and will be formally accepted for publication once it meets all outstanding technical requirements.

Kind regards,

Annamaria Bevivino

Academic Editor

PLOS One

Additional Editor Comments (optional):

Dear Jiraphan Premsuriya,

Thank you for submitting the revised version of your manuscript.

The revisions have been carefully evaluated, and all reviewers’ and editor’s comments have been satisfactorily addressed and incorporated into the manuscript.

The manuscript is now suitable for publication in its present form.

We appreciate your efforts in improving the quality and clarity of the work and thank you for choosing our journal for your submission.

Best regards

Annamaria Bevivino

Academic editor PLOS One
---

## [Editor Report · Acceptance letter]

PONE-D-25-54603R1

PLOS One

Dear Dr. Premsuriya,

I'm pleased to inform you that your manuscript has been deemed suitable for publication in PLOS One. Congratulations! Your manuscript is now being handed over to our production team.

Kind regards,

on behalf of

Prof. Dr. Annamaria Bevivino

Academic Editor

PLOS One